

# Epidemiological characteristics of invasive meningococcal disease and carriage prevalence of *Neisseria meningitidis* in the Xinjiang Uygur Autonomous Region, China, 2004–2023: a retrospective study

Halimubieke Nazhaerbieke[1], Wenhui Fu[2], Zhaoguo Lan[1], Yesitai Muheiyati[1], Tian Tian[1], Caipilima Wuqierjiafu[1] and Na Xie[2]

[1] Department of Epidemiology and Biostatistics, Xinjiang Medical University, Urumqi, Xinjiang Uygur Autonomous Region, China
[2] Xinjiang Uygur Autonomous Region Center for Disease Control and Prevention, Urumqi, Xinjiang Uygur Autonomous Region, China

Corresponding author
Na Xie, xiena371@163.com

## ABSTRACT

**Objective:** The Xinjiang Uygur Autonomous Region (Xinjiang) currently exhibits the high incidence rates of invasive meningococcal disease (IMD) in China. Current epidemiological data on meningococcal carriage rates among asymptomatic individuals remain sparse, with limited population-based studies systematically investigating the prevalence of *Neisseria meningitidis* colonization in the general population. This study employs a dual-pronged epidemiological approach to systematically characterize the clinical-epidemiological profile of IMD in Xinjiang and quantify the nasopharyngeal carriage prevalence of *N. meningitidis* among asymptomatic populations, while identifying region-specific sociodemographic and behavioral determinants of carriage dynamics.

**Methods:** Epidemiological characteristics were analyzed using descriptive epidemiological methods. *N. meningitidis* strains isolated from asymptomatic carriers underwent serogroup characterization *via* multiplex real-time polymerase chain reaction (real-time PCR) targeting conserved capsular biosynthesis loci (serogroups A, B, C, W, X, and Y), with reaction conditions optimized per WHO standardized protocols for meningococcal molecular typing. The $\chi^2$ test was used to compare the *N. meningitidis* carriage rates. A multivariable logistic regression model was used to analyze the risk factors associated with the carriage of *N. meningitidis*.

**Results:** From 2004 to 2023, 1,100 cases of IMD were reported in Xinjiang, with the annual incidence rate fluctuating between 0.00/100,000 and 1.15/100,000 per year. The peak incidence occurred from February to May. The incidence was primarily concentrated in individuals under 20 years old (80.36%). Among 3,075 oropharyngeal swab specimens analyzed, 411 (13.37%; 95% CI [12.18–14.62]) yielded culture-confirmed *N. meningitidis* isolates. Serogroup B (168, 40.88%) emerged as the predominant meningococcal serogroup. Binomial multivariate logistic regression analyses indicated significant associations between the carriage rate of *N. meningitidis* and age, sex, year, region, and vaccination history ($p < 0.05$).

**Conclusion:** Epidemiological surveillance data from 2004 to 2023 revealed a significant decline in the incidence of IMD across Xinjiang, with the epidemiological

profile transitioning from cyclic epidemic peaks to sporadic case clusters. Despite this decline, the carriage rate of *N. meningitidis* remained at elevated levels among healthy populations. The risk of carrying *N. meningitidis* was relatively high among the healthy population in the southern region of Xinjiang, people aged over 16, and those without a vaccination history. Strengthening IMD surveillance in high-risk areas is essential to prevent future outbreaks.

## INTRODUCTION

Invasive meningococcal disease (IMD) is a systemic bacterial infection caused by the gram-negative bacterium *Neisseria meningitidis*, leading to septicaemia and meningitis (*Dubey et al., 2022*; *Read, 2019*). It exhibits pronounced seasonality with bimodal incidence peaks clustered in late winter and early spring, consistent with respiratory pathogen transmission dynamics under cold-dry climatic conditions. Globally, the case fatality rate of IMD is as high as 20%, with a high disability rate and severe disease burden (*Parikh et al., 2020*; *Wang et al., 2019*). It is estimated that there are approximately 1.2 million cases of *N. meningitidis* infection worldwide each year (mainly IMD), with a high death toll of 135,000 (*Jafri et al., 2013*; *Peterson et al., 2019a*), and 5–15% of cases will still have serious sequelae after healing (*Howitz et al., 2009*). *N. meningitidis*, an opportunistic pathogen, and asymptomatically colonizes the nasopharyngeal mucosa in healthy individuals. Humans are the only hosts of this bacterium, and approximately 10% of the healthy population carries *N. meningitidis*, which has become the main source of infection in the current epidemic of IMD (*Alemayehu, Mekasha & Abebe, 2017*; *Wang et al., 2019*). According to the different antigen structures of capsule polysaccharides, *N. meningitidis* can be divided into 12 serogroups: A, B, C, X, Y, W, Z, E, L, H, I, and K. More than 95% of the cases were caused by six serogroups A, B, C, Y, W and X (*Beebeejaun et al., 2020*).

While global surveillance data indicate that IMD incidence has reached historically low levels since the introduction of conjugate vaccines, persistent disparities emerge across demographic and geographic strata and hyperendemic transmission persists in meningitis belt regions and marginalized populations with <50% MenACWY vaccine coverage (*Alderson et al., 2022*; *GBD 2016 Meningitis Collaborators, 2018*). After 2000, the incidence rate of IMD in European and American countries and regions remained at 0.11/100,000 to 2.00/100,000 (*Zhang et al., 2016*). In 2020, the incidence rate will be 0.56/100,000 in Spain and 0.17/100,000 in Brazil (*Alderson et al., 2022*). In 2010, after the introduction of meningococcal polysaccharide conjugate vaccine group A (MPCV-A) in Africa and the start of vaccination, the incidence rate of IMD group A decreased significantly to 0.02/100,000 (*Clément et al., 2015*). Since 2005, China has entered a low incidence period of

IMD. From 2015 to 2019, the incidence rate of IMD in China was 0.078/100,000 people, and the case fatality rate was 11.82% (*Li et al., 2019*). Despite China's national IMD incidence remaining below since 2010, Xinjiang Uygur Autonomous Region (Xinjiang) persists as the nation's hyperendemic epicenter. The average annual incidence rate from 2005 to 2010 was 0.56/100,000, which is significantly higher than that in other provinces; additionally, there were occasional outbreaks and epidemics (*Li et al., 2015*).

Currently, there are many studies on *N. meningitidis* carriage status around the world, molecular epidemiological surveillance reveals marked temporal shifts in meningococcal carriage prevalence among asymptomatic populations (*Kristinsdottir et al., 2023*; *Miellet et al., 2021*; *Ong et al., 2023*). In China, several provinces and cities have carried out surveys on *N. meningitidis* carriage rates (*He et al., 2020*; *Yue et al., 2022*). In contrast, there is less basic information on the *N. meningitidis* carriage status among asymptomatic populations in Xinjiang.

This population-based study was conducted to delineate the epidemiologic characteristics of IMD and carrier status of meningococci (not IMD) in Xinjiang, to establish an evidence-based precision prevention framework of IMD in Xinjiang.

## MATERIALS AND METHODS

### Research type

This population-based retrospective analysis delineates the epidemiology and demographic risk stratification of invasive meningococcal disease (IMD) in Xinjiang Uygur Autonomous Region (Xinjiang) from 2004 to 2023. Additionally, cross-sectional surveys were conducted among healthy populations in Xinjiang to investigate the carriage of *N. meningitidis* in 2016, 2018, 2019, 2022, and 2023.

### Epidemiological characteristics

The data on the incidence of IMD in Xinjiang from 2004 to 2023 were collected from the China Disease Prevention and Control Information System, a comprehensive national database that tracks infectious diseases across the country. Case definitions for IMD were based on the National Invasive Meningococcal Disease Surveillance Program and the Diagnosis of Invasive Meningococcal Disease (WS295-2019), which are standardized protocols established by the Chinese health authorities. These guidelines ensure consistency in the identification and reporting of IMD cases across different regions and healthcare facilities. According to these criteria, cases included both clinically diagnosed and laboratory-confirmed instances of IMD. Clinically diagnosed cases were identified based on symptoms and epidemiological evidence, while laboratory-confirmed cases required positive results from tests such as bacterial culture, real-time polymerase chain reaction (real-time PCR), or antigen detection. Suspected cases, which lacked sufficient clinical or laboratory evidence, were excluded from the analysis to maintain the accuracy and reliability of the data. The demographic data were sourced from the Basic Information System for Disease Control and Prevention of China.

## Survey on the carrier rate of *N. meningitidis* among healthy individuals
### *Study design and sample collection*

According to the incidence of IMD in Xinjiang and its geographic distribution, cross-sectional investigations on the carriage rate of *N. meningitidis* were conducted in the southern and northern regions of Xinjiang in 2016, 2018, 2019, 2022, and 2023. Southern Xinjiang included Aksu Prefecture, Kashgar Prefecture, Hotan Prefecture, and Kizilsu Kirgiz Autonomous Prefecture, with a total of 1,574 participants surveyed. Northern Xinjiang included Ili Kazak Autonomous Prefecture, Tacheng Prefecture, Urumqi City, Karamay City, and Turpan City, with a total of 1,501 participants surveyed. Using random cluster sampling, the population was divided into six age groups: <3, 3–5, 6–10, 11–15, 16–20, and ≥21 years old. At least 40 subjects were sampled from each age group, and 250 subjects were surveyed in each sampling unit. The male-to-female ratio was kept balanced. All participants volunteered and provided written informed consent; for minors or individuals unable to provide consent independently, written consent was obtained from their legal guardians, and assent was sought from the participants themselves where appropriate. Oropharyngeal swab samples were collected by trained staff who had undergone standardized training. A total of 3,075 oropharyngeal swab samples were collected.

The collected oropharyngeal swab samples were inoculated on-site onto a chocolate agar medium supplemented with antibiotics (containing 4.2 mg/L polymyxin B and 3 mg/L vancomycin, produced by Guangdong Huankai Microbial Sci. & Tech. Co., Ltd., Guangzhou, China). After all samples were collected, the plates were transported to the laboratory.

### *Cultivation and identification of N. meningitidis*

After delivery to the laboratory, the chocolate dual-antibiotic medium was incubated at 37 °C in 5% $CO_2$ incubator for 18 to 24 h. Then, single suspicious colony which was round, smooth, moist, convex in the center, with clear edges, colorless or gray and opaque was selected and inoculated onto Columbia blood agar plates (Guangdong Huankai Microbial Sci. & Tech. Co., Ltd., Guangzhou, China). Following 18–24-h incubation on Columbia blood agar plates (5% $CO_2$, 37 °C), a single isolated colony was aseptically harvested using sterile 10 μL loops, subjected to genomic DNA extraction. The universal gene capsular transport gene A (*ctrA*) and the superoxide dismutase C gene (*sodC*) were amplified. The target gene for *N. meningitidis* is *ctrA*. A positive result for *ctrA* can be considered as evidence that the corresponding surveyed population carries *N. meningitidis*. However, a negative result does not rule out the possibility of carrying *N. meningitidis*, as some strains may lack the *ctrA* gene or have mutations in the *ctrA* gene, leading to false-negative results in Real-time PCR. This can be complemented by the detection of the *sodC* gene. If *ctrA* and *sodC* are both positive, the likelihood of it being *N. meningitidis* increases. If *ctrA* is negative but *sodC* is positive, the possibility of a false positive should be considered, and further testing is required. Finally, a colony is confirmed as *N. meningitidis* positive; then, serogroup-specific genes are amplified using the *N. meningitidis* (A, B, C types) and (X, Y, W types) triple nucleic acid detection kits (Shenzhen Shengke Yuan Biotechnology Co.,

Ltd., Shenzhen, China) for serogroup identification. Strains that test negative for all six serogroups are defined as nongenogroup strains. The cycle threshold (Ct) for Real-time PCR amplification is set according to the kit instructions. All experimental procedures, as well as PCR primers and probes, refer to the "Laboratory Methods for the Diagnosis of Meningitis caused by *Neisseria meningitidis*, *Streptococcus pneumoniae*, and *Haemophilus influenzae*, http://whqlibdoc.who.int/hq/1999/WHO_CDS_CSR_EDC_99.7*".

### Ethical approval

Ethics approval for this study was obtained from the Medical Ethics Committee of Xinjiang Uygur Autonomous Region Center for Disease Control and Prevention (No. 2022-001). All participants volunteered and provided written informed consent; for minors or individuals unable to provide consent independently, written consent was obtained from their legal guardians, and assent was sought from the participants themselves where appropriate.

### Statistical analysis

Microsoft Excel 2021 software was used to establish a database to import and organize the data, and the data were processed and statistically analyzed by applying SPSS 26.0 software to analyze the carrying status of the healthy population and characteristics of the distribution of flora. Qualitative data were expressed as frequencies ($n$) and percentages (%). Comparisons of the positivity rates of *N. meningitidis* among different regions, sexes, age groups, years and immunization histories were performed using the $\chi^2$ test, with a $p$ value <0.05 indicating a statistically significant difference. A binomial multivariate logistic regression model was used to analyze the factors influencing the rate of *N. meningitidis* carriage.

## RESULTS

### Epidemiological characteristics

#### Time distribution

From 2004 to 2023, a total of 1,100 cases of IMD were reported in Xinjiang, and the annual incidence rate fluctuated between 0.00/100,000–1.15/100,000. Among them, with the highest incidence rate (1.15/100,000) occurring in 2007. Two peaks occurred in 2009 and 2014, with incidence rates of 0.52 per 100,000 and 0.38 per 100,000, respectively. After 2014, the IMD incidence fluctuated at a low level, with a small peak in 2016, with an incidence rate of 0.19/100,000. Since 2019, the incidence has shown a significant downward trend, with the lowest incidence (0.01/100,000) in 2020, with three cases, and no cases reported in 2021 (Fig. 1A).

From 2004 to 2023, a total of 102 IMD deaths were reported in Xinjiang, with a fatality rate of 9.27%. The annual mortality rate fluctuated between 0.00/100,000–0.09/100,000, and the annual case fatality rate was 0–20%. The death rate in 2007 (0.09/100 000) was the highest, with 19 deaths (8.02%), followed by 2009 (0.05/100 000), with 15 deaths (12.71%). Since 2010, there has been a downward trend in deaths, with a small peak in 2016, and no deaths have been reported since 2020 (Fig. 1B).

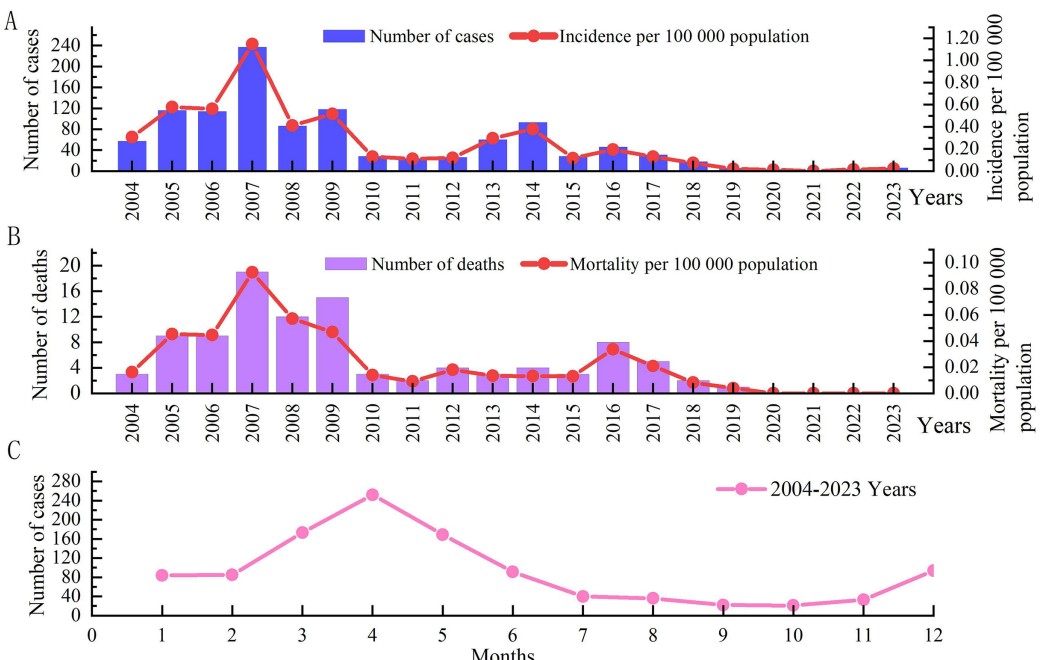

**Figure 1 Time distribution of IMD morbidity and mortality data in Xinjiang, China, from 2004 to 2023.** (A) The histogram shows the number of cases in different years, and the linear graph shows the incidence in different years. (B) The histogram shows the number of deaths in different years, and the linear graph shows the death rate in different years. (C) Shows the seasonal distribution.

The incidence shows obvious seasonality, with a peak from February to May, accounting for 61.73% (679/1,100) of the total incidence. Among them, April had the highest incidence, accounting for 22.91% (252/1,100) of the total incidence. The number of cases in September and October was the lowest, with 22 and 21 cases, respectively, showing an increasing trend from November to December (Fig. 1C).

### Regional distribution

From 2004 to 2023, cases of IMD were reported in all 14 prefectures (states and cities) under the jurisdiction of Xinjiang, with the three highest annual average incidence rates being 0.58/100,000 in Hotan Prefecture, 0.46/100,000 in Kashgar Prefecture and 0.37/100,000 in Urumqi city. The bottom three annual average morbidity rates were 0.02/100,000 in Tacheng, 0.03/100,000 in Karamay/Mongolia Autonomous Prefecture of Bortala (Bortala) and 0.04/100,000 in Changji Hui Autonomous Prefecture (Changji). The three highest annual average mortality rates were 0.06/100,000 in Urumqi city, 0.05/100,000 in Turpan city and 0.05/100,000 in the Hotan region. No fatal cases were reported in the three regions of Changji, Bortala and Altay. The three regions with the highest case fatality rates were 50% in Karamay city, 20% in the Tacheng region and 17.65% in Urumqi city. Of the total number of reported cases, 66.45% (731/1,100) were reported in the southern Xinjiang region, and 33.55% (369/1,100) were reported in the northern Xinjiang region (Fig. 2).

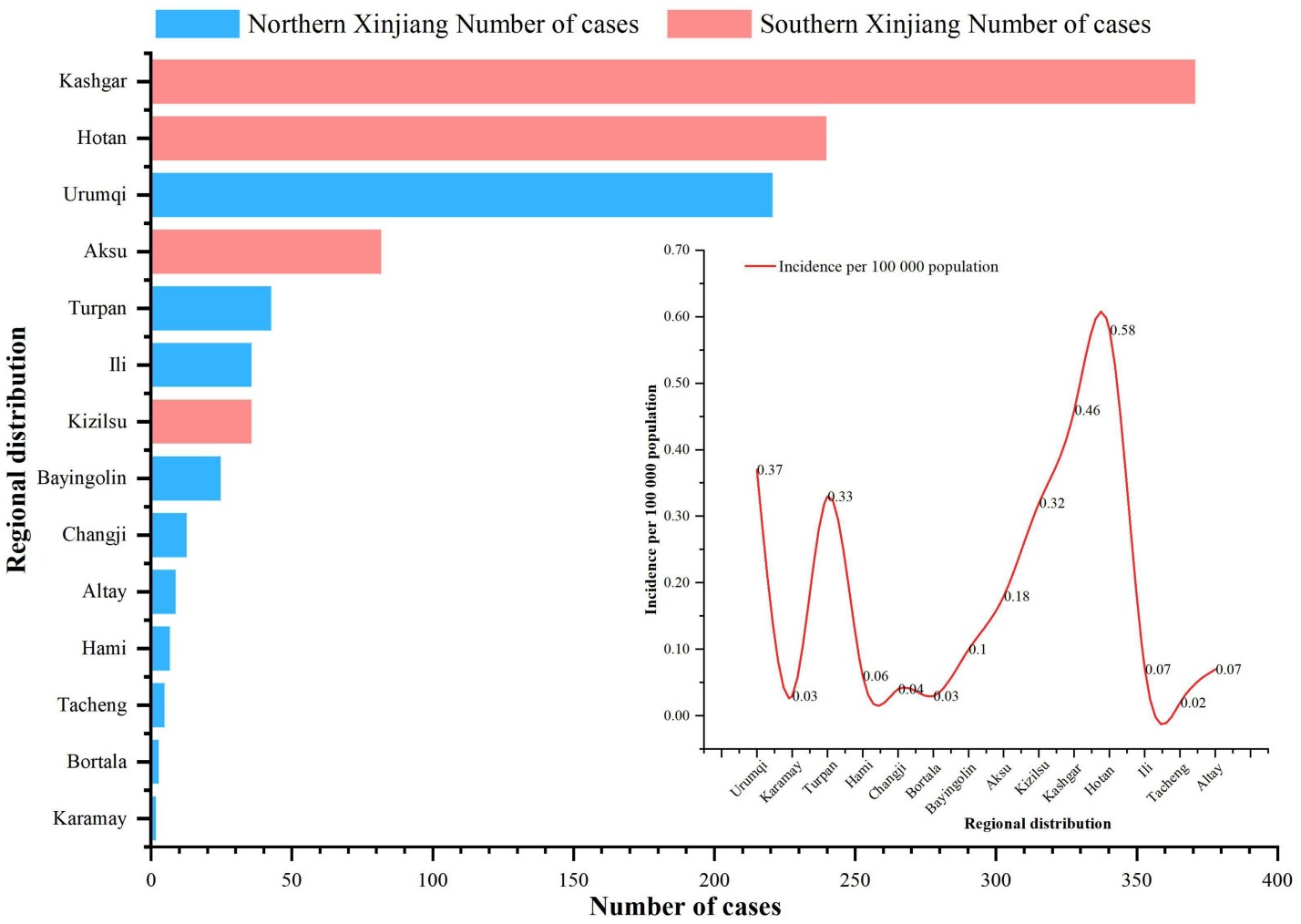

**Figure 2  Regional distribution of IMD incidence data in Xinjiang, China, 2004–2023.** Histogram indicates the number of incidence in different regions (red indicates the southern Xinjiang region, blue indicates the northern Xinjiang region), and the linear graph indicates the incidence rate in different regions.                                                              

### Population distribution

Of the 1,100 cases of IMD reported in Xinjiang from 2004 to 2023, 687 were male and 413 were female, with a male-to-female sex ratio of 1.66:1 (Fig. 3).

From 2004 to 2023, the incidence population in Xinjiang was mainly concentrated in people under 20 years of age, accounting for 80.36% (884/1,100) of the total number of cases. 10–15 years of age had the highest number of incidence cases, accounting for 16.09% (177/1,100) of the total number of cases, followed by 15–20 years of age, accounting for 15.82% (174/1,100) of the total number of cases. The number of reported cases in the age group above 25 years showed a decreasing trend with increasing age (Fig. 3).

The top three cases were scattered children, students and farmers, accounting for 34.18% (376/1,100), 33.27% (366/1,100) and 17.00% (187/1,100) of the total number of

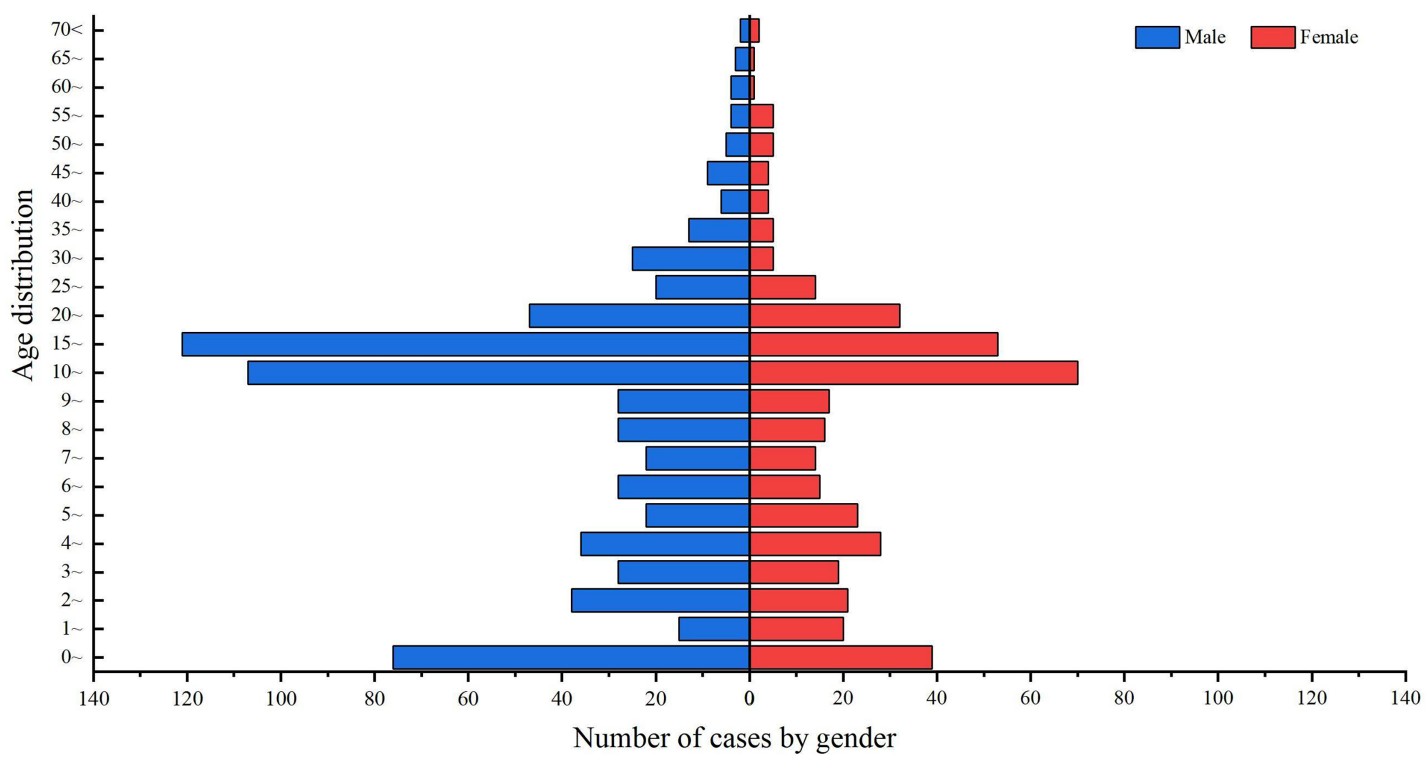

**Figure 3 Age and sex distribution of IMD incidence data in Xinjiang, China, 2004–2023.** Blue indicates males, red indicates females, and the length of the histogram indicates the number of cases of IMD in males and females in each age group.

cases, respectively. Teachers, laborers and businessmen were the last three with the lowest number of cases, with two, five and five cases, respectively (Fig. 4).

## Survey on the carrier rate of *N. meningitidis* among healthy individuals
### Distribution of N. meningitidis carriage and serogroups

A total of 3,075 oropharyngeal swab samples were collected from healthy people, and 411 *N. meningitidis* strains were detected, for an overall carriage rate of 13.37%. The major serogroup was group B, with 168 strains, more than one-third of the total number of strains. This was followed by 158 strains from the nongenogroup and 44 strains from group C, with proportions of 38.44% and 10.71%, respectively. The difference in the carriage rate between the six age groups was statistically significant ($\chi^2 = 76.954$, $p < 0.001$), with the highest rate of 21.34% in the ≥21 years age group, followed by the 16–20 age group (18.93%). Multiple comparisons based on the $\chi^2$ test using the Bonferroni method revealed that the ≥21 years age group and 16–20 years age group were significantly different from all other age groups. The *N. meningitidis* positivity rates of the male and female populations were 15.14% and 11.75%, the difference in the carrier rate between the sexes was statistically significant ($\chi^2 = 7.644$, $p < 0.05$). The difference in *N. meningitidis* carriage rates among asymptomatic carriers in Xinjiang in different years was statistically significant ($\chi^2 = 210.094$, $p < 0.001$), with the highest rate of 25.45% in 2018 and the lowest rate of 3.04% in 2023. Further multiple comparisons were performed, and there was a

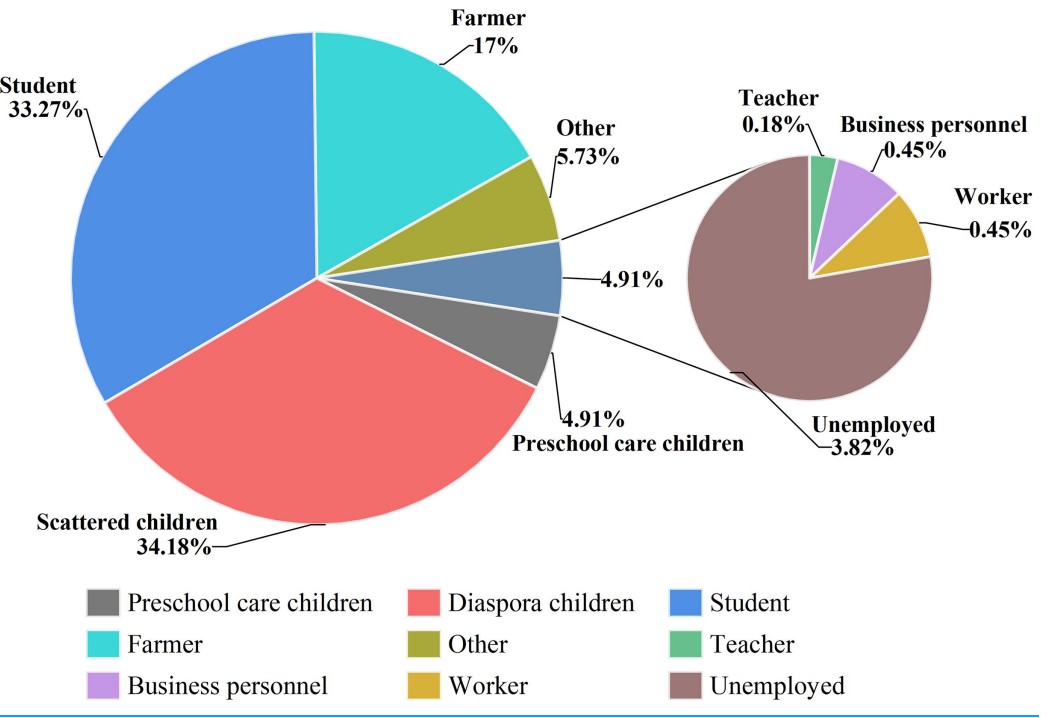

**Figure 4 Occupational distribution of IMD incidence data in Xinjiang, China, 2004–2023.** Different colors indicate the proportion of IMD cases in different occupational groups.

significant difference between 2016, 2018 and 2019 and between 2022 and 2023 ($p < 0.05$). The *N. meningitidis* positivity rates of healthy populations in the southern and northern Xinjiang regions were 19.82% and 6.60%, respectively, and the differences in the carriage rates of populations in different regions were statistically significant ($\chi^2 = 116.077$, $p < 0.001$). The carrier rate of different doses of vaccination in the healthy population in Xinjiang was also different; the highest carrier rate was found in the healthy population with no history of immunization (18.78%), followed by the healthy population with an unknown history of immunization (15.89%). The lowest carrier rate was found in the ≥3-dose vaccination population (7.83%), and the difference was statistically significant ($\chi^2 = 52.834$, $p < 0.001$). Multiple comparisons revealed a statistically significant difference between the healthy population with no history of immunization and those who were vaccinated with every dose ($p < 0.05$) (Table 1).

### Risk factors for N. meningitidis carriage

After binomial multivariate logistic regression analyses, age, sex, year and region were the risk factors influencing the rate of *N. meningitidis* carriage ($p < 0.05$) (Table 2). Among the different age groups, the risk of *N. meningitidis* carriage generally tended to increase and then decrease with age. Individuals aged 16–20 years (odds ratio (OR) 4.402, confidence interval (CI) [2.595–7.469], $p < 0.001$) and those aged ≥21 years (OR 2.853, CI [1.679–4.847], $p < 0.001$) had a relatively high risk of carrying *N. meningitidis*, while those aged from 0 to 10 years had a relatively low risk. Males (OR 1.293, CI [1.033–1.619],

**Table 1  *N. meningitidis* carriage status in healthy people in Xinjiang, China.**

| Index | Number | Number (%) of *N.meningitidis* Carriage | Serum group distribution | | | | | | |
|---|---|---|---|---|---|---|---|---|---|
| | | | A | B | C | X | Y | W | Nongenogroup[g] |
| Age group | | | | | | | | | |
| <3 | 402 | 32 (7.96) | 0 | 13 | 2 | 0 | 3 | 0 | 14 |
| 3–5 | 366 | 32 (8.74) | 0 | 7 | 4 | 0 | 3 | 2 | 16 |
| 6–10 | 509 | 44 (8.64) | 0 | 20 | 3 | 2 | 3 | 0 | 16 |
| 11–15 | 659 | 73 (11.08) | 0 | 29 | 4 | 4 | 7 | 0 | 29 |
| 16–20 | 544 | 103[ac] (18.93) | 0 | 43 | 14 | 3 | 7 | 0 | 36 |
| ≥21 | 595 | 127[bc] (21.34) | 0 | 56 | 17 | 1 | 5 | 1 | 47 |
| Gender | | | | | | | | | |
| Male | 1,466 | 222 (15.14) | 0 | 93 | 32 | 7 | 12 | 1 | 77 |
| Female | 1,609 | 189 (11.75) | 0 | 75 | 12 | 3 | 16 | 2 | 81 |
| Years | | | | | | | | | |
| 2016 | 513 | 84[d] (16.37) | 0 | 25 | 12 | 1 | 6 | 0 | 40 |
| 2018 | 664 | 169[d] (25.45) | 0 | 69 | 18 | 1 | 9 | 2 | 70 |
| 2019 | 702 | 120[d] (17.09) | 0 | 56 | 12 | 7 | 9 | 0 | 36 |
| 2022 | 506 | 17 (3.36) | 0 | 6 | 0 | 1 | 2 | 0 | 8 |
| 2023 | 690 | 21 (3.04) | 0 | 12 | 2 | 0 | 2 | 1 | 4 |
| Region | | | | | | | | | |
| South Xinjiang | 1,574 | 312 (19.82) | 0 | 138 | 39 | 9 | 24 | 0 | 102 |
| Northern Xinjiang | 1,501 | 99 (6.60) | 0 | 30 | 5 | 1 | 4 | 3 | 56 |
| Vaccination history | | | | | | | | | |
| Not vaccinated | 735 | 138[e] (18.78) | 0 | 50 | 26 | 8 | 7 | 1 | 46 |
| Men A 1 dose | 199 | 18 (9.05) | 0 | 6 | 2 | 0 | 1 | 0 | 9 |
| Men A 2 dose | 220 | 28 (12.73) | 0 | 19 | 3 | 0 | 2 | 0 | 4 |
| Men AC ≥3 dose | 971 | 76 (7.83) | 0 | 33 | 0 | 0 | 8 | 2 | 33 |
| Not quite clear | 950 | 151[f] (15.89) | 0 | 60 | 13 | 2 | 10 | 0 | 66 |
| Total | 3,075 | 411 (13.37) | 0 | 168 | 44 | 10 | 28 | 3 | 158 |

**Notes:**
Multiple comparisons of the results revealed the following: 16–20 age group compared with the first four age groups.
[a] *p* < 0.001; ≥21 age group compared with the first four age groups.
[b] *p* < 0.001; 16–20 age group compared with the ≥21 age group.
[c] *p* > 0.05; comparison with 2022 and 2023.
[d] *p* < 0.05; comparison with the population vaccinated with every dose.
[e] *p* < 0.05; and comparison with the 1-dose and ≥3-dose populations.
[f] *p* < 0.05.
[g] Strains that test negative for all six serogroups are defined as nongenogroup strains.

*p* < 0.05) had a greater risk of carrying *N. meningitidis* than females. The risk of carrying *N. meningitidis* was highest in 2018 (OR 8.601, CI [5.149–14.365], *p* < 0.001), while in 2022 (OR 1.128, CI [0.572–2.223], *p* > 0.05) the risk of carrying *N. meningitidis* was relatively low compared with other years. The risk of *N. meningitidis* carriage among asymptomatic carriers in the southern Xinjiang region (OR 1.831, CI [1.350–2.484], *p* < 0.001) was significantly greater than that in the northern Xinjiang region. In terms of vaccination history, the risk of carrying *N. meningitidis* increased in asymptomatic carriers with no

**Table 2 Binomial multivariate logistic regression analysis of influencing factors of the *N. meningitidis* carrier rate in the healthy population in Xinjiang, China.**

| Index | OR | 95% CI | *p*-value |
|---|---|---|---|
| Age group | | | |
| <3 | Ref | | |
| 3–5 | 1.344 | [0.756–2.386] | 0.314 |
| 6–10 | 1.516 | [0.879–2.615] | 0.135 |
| 11–15 | 2.347 | [1.392–3.956] | 0.001 |
| 16–20 | 4.402 | [2.595–7.469] | <0.001 |
| ≥21 | 2.853 | [1.679–4.847] | <0.001 |
| Gender | | | |
| Male | 1.293 | [1.033–1.619] | 0.025 |
| Female | Ref | | |
| Years | | | |
| 2016 | 6.788 | [4.034–11.424] | <0.001 |
| 2018 | 8.601 | [5.149–14.365] | <0.001 |
| 2019 | 4.728 | [2.801–7.981] | <0.001 |
| 2022 | 1.128 | [0.572–2.223] | 0.729 |
| 2023 | Ref | | |
| Region | | | |
| South Xinjiang | 1.831 | [1.350–2.484] | <0.001 |
| Northern Xinjiang | Ref | | |
| Vaccination history | | | |
| Not vaccinated | 2.324 | [1.384–3.904] | 0.001 |
| Men A 1 dose | Ref | | |
| Men A 2 dose | 1.466 | [0.784–2.742] | 0.231 |
| Men AC ≥3 dose | 0.854 | [0.499–1.462] | 0.565 |
| Not quite clear | 1.900 | [1.136–3.179] | 0.014 |

Note:
The final binomial multivariate logistic regression model included age group, gender, year, region, and vaccination status.
Abbreviations: CI is the confidence interval; OR is the odds ratio; *p* is the probability value; Ref is the reference value.

history of immunization (OR 2.324, CI [1.384–3.904], $p < 0.01$) or with an unknown vaccination history (OR 1.90, CI [1.136–3.179], $p < 0.01$) (Table 2).

## DISCUSSION

In recent years, the incidence of IMD in Xinjiang has reached a relatively high level in the country, and some areas have even experienced an explosive epidemic (*Li et al., 2015*). Therefore, this study investigated the epidemiological characteristics of IMD in Xinjiang from 2004 to 2023 and the carriage situation of *N. meningitidis* among the healthy population. Epidemiological transition analysis revealed that Xinjiang Uygur Autonomous Region (Xinjiang)—China's meningitis hyperendemic frontier bordering Central Asia— underwent a paradigm shift from cyclical epidemic patterns to sustained hypoendemic transmission, yet still maintaining higher disease burden compared to Europe and the

United States (*Parikh et al., 2020*; *Sohn et al., 2022*). Since the 1980s, China has successfully developed a polysaccharide vaccine for group A IMD and has conducted extensive vaccination, resulting in a decreasing trend in the incidence of IMD annually (*Shao et al., 2006*). However, epidemiological surveillance data reveal persistent reporting of laboratory-confirmed IMD cases across Xinjiang, and this study revealed that these cases were mainly distributed in southern Xinjiang (Hotan and Kashgar), which was consistent with the results of carrier rate among healthy individuals. Based on practical considerations, the economic development level of northern Xinjiang was significantly higher than southern Xinjiang (*Abulimiti et al., 2022*). Beyond socioeconomic determinants and healthcare access barriers, regional climate characteristics and vaccination conditions were also the main factors (*Xie et al., 2021*). The southern Xinjiang region, characterized by a temperate continental arid climate with annual precipitation <100 mm, experiences frequent dust-laden winds that exacerbate respiratory pathogen transmission. Therefore, in addition to improving medical facilities and strengthening public education, it is necessary to enhance laboratory monitoring capabilities and facilitate cooperation between medical institutions and disease control departments to establish a monitoring system for IMD cases. Given Xinjiang's climate, targeted prevention and control measures should be implemented during the peak incidence seasons of winter and spring, along with initiatives to popularize personal protection knowledge, effectively reducing the transmission risk of respiratory pathogens such as those causing IMD.

In the current study, we found that the seasonal peak of IMD cases in Xinjiang was from February to May, with the highest number of cases occurring in April, which was similar to findings in other Chinese provinces (*Chen et al., 2018*; *Li et al., 2015*). This may be related to climatic factors, for example, during the peak season of influenza in China, there is a seasonal difference between the north (January) and the south (June to July), and the climate in the south is warm and humid (*Shu et al., 2010*), which also suggests that the climate has a certain influence on the prevalence of infectious diseases. The Xinjiang region is in the northwestern region of China, and the climate is different from other regions. The climate in the southern Xinjiang region is an extreme continental climate of the tropics, with hot and dry summers, low rainfall, and abundant sunshine. The results of the study by *Jusot et al. (2017)* showed that high temperatures and low visibility were identified as important risk factors for bacterial meningitis. Therefore, climate surveillance is used to predict the prevalence of IMD, in order to prevent and control outbreaks of IMD. In addition to climatic factors, national social factors should also be considered. China's Spring Festival migration (January–March) represents the planet's largest annual human migration event, with >200 million interregional travelers generating transient population density increases of 3–5 fold in receiving communities (*Wang et al., 2013*). Mass population mobility during holiday periods generates two synergistic risk amplifiers for meningococcal transmission: prolonged close contact (<1 m interpersonal distance for >4 h) in overcrowded transport nodes (airports/railway stations reaching 6–8 persons/m$^2$), and aerosol dispersion efficiency increases in enclosed vehicles, these findings underscore the critical need for time-sensitive meningococcal prevention strategies during mass migration periods.

The results of many studies showed that cases of IMD have been reported in people of all age groups, mainly occurring among those under 15 years old (*Flountzi et al., 2019*; *Li et al., 2018*; *Yue et al., 2022*). From 2004 to 2023, the reported cases of IMD in Xinjiang were mainly in people under 20 years of age, with the highest number of cases occurring in the 10–15-year-old age group, which was similar to the results of this healthy population carrier rate survey. The highest carrier rate was found in the ≥21 years age group, followed by the 16–20 years age group. Some researchers had shown that 40% of *N. meningitidis* usually colonize the nasopharynx of adults and occasionally cause IMD disease (*Borrow, 2012*). Adolescents aged 15–20 years—primarily school and university students—represent a high-risk demographic for IMD, which is more likely to lead to aggregated IMD outbreaks. This has been similarly reported in many investigations (*Borrow et al., 2017*; *Coronell-Rodriguez et al., 2023*; *McMillan et al., 2024*; *Peterson et al., 2019b*; *Tefera et al., 2020*). In particular, students in group accommodations at school are at high risk of contracting *N. meningitidis* (*Miellet et al., 2021*). Therefore, the prevention and control of IMD in schools should be given high priority. It is crucial to remain vigilant about the transmission of carriers among students, strengthen sanitation and cleaning in classrooms and dormitories, and conduct targeted catch-up vaccination campaigns for IMD to prevent outbreaks on campus.

The *N. meningitidis* carrier status of healthy people is closely related to the prevalence of IMD epidemics (*The MenAfriCar Consortium, 2015*; *Steurer et al., 2020*). Therefore, it is important to investigate the *N. meningitidis* carrier status of healthy populations for the prediction of IMD epidemics and the formulation of immunization strategies. In the present study, a survey of *N. meningitidis* carriage status was carried out among 3,075 healthy people in Xinjiang, and 411 *N. meningitidis* strains were detected, with an overall carriage rate of 13.37%, which was significantly greater than the upper respiratory *N. meningitidis* carriage status of university students in Kampar, Malaysia, which was 5% (*Ong et al., 2023*); and in Iceland from 2019 to 2021, which was 6.50% (*Kristinsdottir et al., 2023*) and mainly involved adolescent groups. This may be related to the vaccination situation. In some countries, vaccination is provided to teenagers to reduce the carriage rate and the incidence of IMD among the whole population. The available vaccines differ across the countries, including both conjugate vaccines and polysaccharide vaccines, but the use of only conjugate vaccines could interrupt the acquisition of carriage (*Bai et al., 2019*). Currently, countries such as the United States and the United Kingdom have implemented the MenACWY conjugate vaccine in adolescent populations, resulting in effective control of both *N. meningitidis* carriage rates and the incidence of IMD (*Dretler, Rouphael & Stephens, 2018*; *Vetter et al., 2016*). However, in most countries, *N. meningitidis* vaccines are only provided to high-risk groups. In China, only polysaccharide vaccines are included in the expanded programme on immunisation. Given the increasing carriage rate among the adolescent population, it is important for China to consider following the example of other countries and modifying the immunisation strategy to use conjugate vaccine against IMD-related serogroups in teenagers.

The use of vaccines has a significant effect on controlling the prevalence and distribution of IMD diseases (*Pizza, Bekkat-Berkani & Rappuoli, 2020*). The most

prevalent strains of IMD in China in the past were Group A and Group C. After the 1980s, with the use of group A vaccines, the incidence of group A IMD cases in China showed an overall decreasing trend (*Abio, Neal & Beck, 2013*). The serotype and flora distribution of *N. meningitidis* changed accordingly (*Peterson et al., 2019b*). Recent research in China showed that the proportion of IMD patients with Group A continued to decrease, the number of patients with Group B increased significantly, accounting for more than half of the total number of IMD patients (*Parikh et al., 2020*). This finding was basically consistent with the results of this study. Serogroup B emerged as the predominant *Neisseria meningitidis* carriage strain among asymptomatic populations in Xinjiang. The results of this study were basically consistent with those of *Peterson et al. (2019a)*, *Yue et al. (2022)*, *Sofer-Sali et al. (2022)* and others abroad. Both polysaccharide- and protein-bound vaccines have been developed that can be used against serogroup A, C, W, and Y, with direct and indirect (population) protection, but cannot provide cross-protection against other *N. meningitidis* serogroup (*Borrow, 2012*). However, polyglycoprotein-binding vaccines against group B are difficult to develop, because their structure is similar to that of polysialic acid on human nerve cells, resulting in the low immunogenicity and difficulty in generating effective antibodies. Therefore, the development of such vaccines against group B meningococcal has been challenging (*Ladhani et al., 2020*). At present, some developed countries have obtained licenses for group B meningococcal vaccines (*Borrow, 2012*; *Ladhani et al., 2020*), notably, China presently lacks nationally licensed vaccines targeting *Neisseria meningitidis* serogroup B (MenB) strains. In this study, nongenogroup bacteria accounted for 38.44% of all detected strains. Many studies have shown that nongenogroup strains were carried by healthy people and rarely pathogenic (*Coch Gioia et al., 2015*), but there have been cases of IMD caused by nongenogroup strains in China (*Zhang et al., 2019*). Therefore, the monitoring of nongenogroup *N. meningitidis* should not be ignored, the monitoring and related research of nongenogroup strains should be strengthened, which has important public health significance.

A survey on the rate of *N. meningitidis* infection among healthy people revealed that the rate of *N. meningitidis* infection in other years differed from that in 2022 and 2023. The results were consistent with the incidence data from this study. Since 2019, there has been a significant downward trend in the incidence rate, with the lowest rate recorded in 2020 and no cases reported in 2021. This might be attributed to the impact of the COVID-19 pandemic, people generally took personal protective measures such as wearing masks, reducing going out and gathering, frequent hand washing and ventilation. This effectively reduced the possible transmission of respiratory infectious diseases such as IMD (*Alderson et al., 2022*; *Liu et al., 2021*).

In this study, relevant factors such as age, sex, year, region and vaccination status were significantly correlated with *N. meningitidis* carrier status. These factors were significantly related to risk factors according to different studies (*Coch Gioia et al., 2015*; *Ong et al., 2023*; *Tefera et al., 2020*). There was a significant positive correlation between *N. meningitidis* carriage prevalence in healthy populations and IMD incidence rates.

This study has several limitations. Firstly, there is a potential risk of bias in vaccination information among healthy adult populations. For older age groups, vaccination records

lack systematic electronic archiving, and their immunization history was primarily collected through personal recall or family oral descriptions. To address this issue, future research could employ multi-source data verification and supplement biological indicators to reduce information bias caused by subjective recollection. Additionally, expanding the inclusion of samples from recently vaccinated populations would enhance analytical precision. Secondly, there are gaps in serogroup distribution characteristics data of case strains. To improve this limitation, subsequent studies will establish a real-time strain sharing mechanism to ensure timely sample collection during case occurrences. Through molecular biology techniques, we will systematically analyze strain serogroup distribution, antimicrobial resistance profiles, and clonal transmission chains. These efforts will accumulate critical data for accurately assessing vaccine protection effectiveness and formulating regional prevention and control strategies.

## CONCLUSION

Longitudinal surveillance (2004–2023) revealed a significant decline in IMD incidence across Xinjiang, transitioning from cyclical epidemic patterns to sustained hypoendemic transmission. Notwithstanding declining IMD incidence, *N. meningitidis* carriage persisted at elevated levels, with multilevel logistic regression identifying key determinants of regional heterogeneity. We recommend expanding the surveillance network for IMD and conducting annual population-based surveys of *N. meningitidis* carriage during early IMD outbreaks. This dual approach will enhance predictive capability for regional IMD trends, enabling timely adjustments to prevention and control strategies—ultimately strengthening evidence-based IMD management in Xinjiang.

### Funding

This work was supported by Xinjiang Tianshan Yingcai Medical and Health Talent Cultivation Program, China: No. TSYC202301B097. Xinjiang Natural Population Cohort Construction and Active Health Innovation Team, China: No. 2022TSYCTD0013. The funders had no role in study design, data collection and analysis, decision to publish, or preparation of the manuscript.

### Grant Disclosures

The following grant information was disclosed by the authors:
Xinjiang Tianshan Yingcai Medical and Health Talent Cultivation Program, China: TSYC202301B097.
Xinjiang Natural Population Cohort Construction and Active Health Innovation Team, China: 2022TSYCTD0013.

### Competing Interests

The authors declare that they have no competing interests.

## Author Contributions

- Halimubieke Nazhaerbieke conceived and designed the experiments, performed the experiments, analyzed the data, prepared figures and/or tables, authored or reviewed drafts of the article, and approved the final draft.
- Wenhui Fu conceived and designed the experiments, performed the experiments, analyzed the data, authored or reviewed drafts of the article, and approved the final draft.
- Zhaoguo Lan conceived and designed the experiments, analyzed the data, prepared figures and/or tables, authored or reviewed drafts of the article, and approved the final draft.
- Yesitai Muheiyati conceived and designed the experiments, performed the experiments, analyzed the data, prepared figures and/or tables, authored or reviewed drafts of the article, and approved the final draft.
- Tian Tian conceived and designed the experiments, analyzed the data, prepared figures and/or tables, and approved the final draft.
- Caipilima Wuqierjiafu conceived and designed the experiments, performed the experiments, analyzed the data, prepared figures and/or tables, and approved the final draft.
- Na Xie conceived and designed the experiments, performed the experiments, analyzed the data, authored or reviewed drafts of the article, and approved the final draft.

## Human Ethics

The following information was supplied relating to ethical approvals (*i.e.*, approving body and any reference numbers):

Ethics approval for this study was obtained from Medical Ethics of Xinjiang Uygur Autonomous Region Center for Disease Control and Prevention (No. 2022-001). Written informed consent was obtained from all subjects and/or their legal guardian(s).

## Data Availability

The raw measurements are available in the supplemental files.

## Supplemental Information

Supplemental information for this article can be found online at http://dx.doi.org/10.7717/peerj.19772#supplemental-information.

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
