# Peer review of "Epidemiological characteristics of invasive meningococcal disease and carriage prevalence of Neisseria meningitidis in the Xinjiang Uygur Autonomous Region, China, 2004–2023: a retrospective study"

_PeerJ, doi:10.7717/peerj.19772_

## Round 0.1 · original submission · Major Revisions

This study offers valuable insights into the transmission dynamics of meningococcal strains, which can enhance control and prevention measures for meningococcal infections. However, the manuscript contains several errors that need significant revision. Extensive language corrections are necessary; it is strongly recommended that the authors seek professional language editing or carefully review the text to ensure clarity and coherence.

Additionally, the authors should conduct a thorough review of their work, comparing it with other publications in the field to improve the clarity and quality of their manuscript. The literature review, data analysis, and interpretation of results should be reassessed to ensure accuracy. Furthermore, the data must be carefully reviewed, and the conclusions should align appropriately with the results obtained.

Reviewer 1 ·

Basic reporting

The paper is generally well written although it could do with a thorough proof reading to weed out some incorrect use of tense and other minor grammatical errors.

The structure and figures are acceptable.


Refer to additional comments.

Experimental design

The experimental design (aim and methods used) is sound, however, some of the methods aren't reflected in the results.


Refer to additional comments.

Validity of the findings

The data is valuable for ongoing epidemiology, although I feel there is a lack of strain characterisation which minimises the impact of the study. I also feel the conclusions drawn by the authors are not wholly supported by the data.

Refer to additional comments.

Additional comments

This paper describes data collected on IMD rates and pharyngeal carriage rates in Xinjiang in China. The paper is generally well written although it could do with a thorough proof reading to weed out some incorrect use of tense and other minor grammatical errors. The data is valuable for ongoing epidemiology, although I feel there is a lack of strain characterisation which minimises the impact of the study. I also feel the conclusions drawn by the authors are not wholly supported by the data.

Major comments

Results: The methods indicate that the IMD strains were characterized for capsular group (genogrouping), but I can’t find any results showing a breakdown of the group distribution among IMD cases.

Conclusions: “IMD in Xinjiang is undergoing an epidemic trend and serogroup transition”. I dont think this is supported by the data. It seems that IMD rates are relatively low at this time (or at least the latest data (2023) suggests this), so I don’t know how we can say its an epidemic. Also there is no serogroup data for IMD strains in this paper so how can you claim there is a serogroup transition underway??

Results: For the pharyngeal carriage survey, the methods suggest both ctrA and sodC were used as targets. ctrA will only detect capsulated meningococci (so is a very conservative measure and won’t represent all meningococcal carriage). SodC is found among almost all meningococcal strains but can also lead to cross-reactions with other genera (e.g. haemophilus). The authors haven’t indicated in the results what the results were using these two targets. The raw data suggest contains results for ctrA, so did the study include sodC? If not, you should acknowledge that the carriage rate is not for ALL meningococci, but only capsulated strains.

Discussion: what vaccine(s) did these people receive? Was it a conjugated vaccine? This would explain the reduction in carriage, but plain polysaccharide vaccines have little effect on carriage.
Throughout: the authors keep referring to IMD as epidemic (e.g. IMD is a global epidemic disease (line 61). Do you mean endemic? Please check your definitions.

Minor comments

Introduction- line 46: IMD is not an acute respiratory infection. Although the main route of transmission is respiratory, the typically symptoms are not respiratory but invasive/ systemic bacterial infection (e.g. sepsis and meningitis).

Methods: do you have a reference for the genogrouping kit used?

Results: In the carriage survey, were there any participants from which >1 strain/serogroup was detected? Please state.

Line 315: What is “IMD vaccine detection and replanting”?

Reviewer 2 ·

Basic reporting

In this manuscript, the authors present a study on the epidemiological characteristics of invasive meningococcal disease (IMD) in the Xinjiang Uygur Autonomous Region in China, as well as the potential risk of Neisseria meningitidis carrier status in healthy people as a cause of IMD. Given that the incidence of IMD is high in the Xinjiang Uygu region, this study is particularly important for understanding the transmission dynamics of meningococcal strains in order to improve control and prevention measures for meningococcal infections.
In my opinion, the title could be clearer since this study focuses on the epidemiological characteristics of invasive meningococcal disease (IMD) and the carrier status of IMD.
In the Introduction section and throughout the manuscript, I noticed some lack of clarity on the part of the authors, as they confuse the meaning of invasive meningococcal disease, since IMD is not an ‘acute respiratory infection’ (line 46), but an acute bacterial infection that can cause septicaemia and meningitis. I suggest that the authors improve the introduction and add references on N. meningitidis carriage studies. The English language throughout the manuscript should be improved in order to reinforce the importance of the research question and allow for better comprehension.
In the Materials and Methods section, I suggest a better description, since the methodologies used are not very clear. In addition, the manuscript has some writing errors and a lack of references to support the results.

Experimental design

This manuscript presents an original work with a defined research question and a wide set of data. However, the Materials and Methods section is not well structured, which leads to confusion. For example, there is insufficient description, particularly in the Pathogen Testing section, of the pharyngeal swab samples used (how many were collected and how they were collected (oropharynx, nasopharynx)). I would also suggest that the authors explain why they use real-time PCR on DNA from pharyngeal swab samples and DNA from strains when the studies state that the identification of Neisseria meningitidis is carried out using DNA from strains.
The authors can find information in a number of studies, such as:
Miellet, W.R., Mariman, R., Pluister, G. et al. Detection of Neisseria meningitidis in saliva and oropharyngeal samples from college students. Sci Rep 11, 23138 (2021). https://doi.org/10.1038/s41598-021-02555-x.
Steurer, LM., Hetzmannseder, M., Willinger, B. et al. Pharyngeal carriage rates of Neisseria meningitidis in health care professionals at a tertiary university pediatric hospital. Eur J Clin Microbiol Infect Dis 39, 1703–1709 (2020). https://doi.org/10.1007/s10096-020-03894-9

Validity of the findings

The conclusion should be revised so that it is well-structured and linked to the aim of the research.

---

## Round 0.2 · Minor Revisions

The authors have addressed the main concerns raised in the reviews. However, the revised manuscript still merits further consideration. Please provide point-by-point responses to the comments made by the Reviewer regarding the new version of your manuscript. I concur with the Reviewer that the discussion could be improved by including an analysis of the study's limitations.

Reviewer 1 ·

Basic reporting

I think the paper is much improved with clear English and formatting. There are a few minor changes to be made (see minor comments)

Experimental design

The experimental method is sound and much clear in this revised manuscript

Validity of the findings

The data is valuable for ongoing epidemiology, although I feel there is a lack of strain characterisation which minimises the impact of the study.

Additional comments

The authors have made substantial improvements to this manuscript. The spelling and grammar is much improved (although I have highlighted some remaining corrections in the minor comments), and the methods and results are much clearer. I feel there are still some small remaining changes needed to bring this up to publication standard.

Major comments:

Discussion: I feel the most significant limitation of these data is the lack of any kind of IMD strain characterisation. Although the authors have provided a detailed epidemiological analysis of the IMD case numbers, without knowing what strains (at least serogroups) are causing disease its very difficult to make any recommendations in terms of vaccination. How do you know whether to develop/implement a MenACWY or a MenB vaccine if you don’t know what strains are causing the disease? You can try and use carriage data as a guide, but there isn’t always a perfect correlation between the strains carried and the strains causing IMD. (e.g. the MenC epidemics in Europe in the 1990s was caused by a cc11 strain that had a low carriage rate). I feel the authors need to acknowledge and address this major limitation in the discussion, and perhaps make any suggestions of ways to assess IMD strain distribution in the region in order to inform a future vaccination strategy.
Line 291: The authors claim strengthening public health and education in the Xinjiang could improve meningococcal vaccination but we don’t have any data on serogroup distribution of the IMD cases. Since the available MenA vaccine will only work against MenA IMD strains, we cant conclude the further use of this vaccine will have any impact on IMD in the region.
Lines 389-402: I’m not sure what the conclusion is here. You say there appears to be a link between meningococcal carriage and vaccination status with fully vaccinated individuals having a lower carriage rate. Are you suggesting the vaccination has reduced the carriage rate? Table 1 indicates that people have received a MenA vaccine (or a MenAC vaccine??). This would only impact MenA (and MenC strains?). None of your participants carried MenA strains. There does appear to be an association with vaccination and MenC carriage but again its not clear if MenC is in the vaccine? Have you controlled for age in your carriage rate vaccine analysis? You show that the carriage rate varies by age, perhaps the vaccinated people are just the adolescents/adults who carry more? And the unvaccinated are younger children who carry less?

Minor comments:

Line 65: AND asymptomatically colonises…
Line 94-95: carrier status of meningococci (not IMD)
Line 148: I would highlight that the majority of carried strains do not harbour ctrA. Only capsulated strains have it, so you when using ctrA for carriage studies, you’re not measuring overall N. meningitidis carriage, but carriage of capsulated/ groupable strains.
Line 290: change “vaccination of meningococcal vaccines” to “meningococcal vaccination”
Line 321: “Some researchers had shown that 40% of N. meningitidis cases usually colonize the nasopharynx”. Not sure what this means. Please rephrase.
Line 358/9: “with Group A” and “with Group B”?
Line 365: Serogroup
Lines 363-367: Evidence suggests that polysaccharide (plain) do not induce indirect/ herd protection (i.e. prevent carriage acquisition).
Line 367: Serogroup not Serotype
Fig 4. Change “Unemployment to unemployed”

---

## Round 0.3 · accepted · Accept

The authors have satisfactorily addressed most of the review comments and made the necessary changes to the manuscript.